# Cutting Compensation in the Time-Frequency Domain for Smeared Spectrum Jamming Suppression

Li Zeng [1] , Hui Chen [2], Zhaojian Zhang [2], Weijian Liu [2] , Yongliang Wang [1,2,*] and Liuliu Ni [2]

1 School of Electronic Information, Wuhan University, Luojia Road, Wuhan 430072, China; ali_youxiang@163.com
2 Wuhan Electronic Information Institute, Huangpu Road, Wuhan 430019, China; chhglr@sina.com (H.C.); zzj554038@163.com (Z.Z.); liuvjian@163.com (W.L.); niliuliu2017@163.com (L.N.)
* Correspondence: ylwangkjld@163.com

**Abstract:** Smeared spectrum (SMSP) jamming is a new type of distance false-target jamming. It consists of multiple sub-pulses, which results in dense false targets at the radar receiver and affects the detection of target signal. Aiming at the suppression of SMSP jamming, in this paper we propose a fast jamming suppression method based on the time-frequency domain according to the time-frequency distribution characteristic of SMSP jamming. This method completely suppresses SMSP jamming in the time-frequency domain, retains the time-frequency points of the remaining target signal, uses the compensation method to obtain the lost target signal, and then restores the time-frequency distribution characteristic of the target signal. It will not produce jamming sidelobe after the recovered signal matched filtering in the time domain. Moreover, we can obtain the Doppler frequency in the time-frequency domain, which can be adopted in practical engineering applications. The simulation results illustrate the effectiveness of the proposed method.

**Keywords:** smeared spectrum (SMSP) jamming; fractional Fourier transform (FRFT); time-frequency domain; linear frequency modulation (LFM) signal; modulation slope

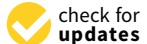


## 1. Introduction

Smeared spectrum (SMSP) jamming is formed by compressing the linear frequency modulation (LFM) signal transmitted by the radar in the time domain, and then repeating it many times to keep the time width of the signal unchanged [1,2]. The SMSP jamming has a high degree of similarity with the target echo, and a large number of dense false targets with a comb structure are formed after pulse compression, which poses a serious threat to radar detection and tracking [3,4]. Therefore, in order to improve the operational efficiency and battlefield survivability of radar, effective suppression of SMSP jamming is urgently needed.

At present, the research on SMSP jamming suppression can be divided into three categories. The first type is to suppress SMSP jamming after separating the target signal and jamming signal. For example, in [5], the difference in polarization information between the jamming signal and target signal was used to propose an anti-SMSP jamming algorithm based on jamming reconstruction and and blind source separation (BSS). Although this method can reduce the calculation work, the estimation of the target polarization state always lagged behind the change of the target polarization state, so it needs further research in the actual environment. According to the form and characteristic of SMSP jamming, the target signal and jamming signal were separated by compressed sensing (CS) theory and a signal processing method against SMSP jamming was proposed in [6], which did not require repeated decoupling of the signal in the time-frequency domain, and had the advantages of fewer iterations and higher computational efficiency. The BSS algorithm was used in [7] to separate the target signal from the mixed signal, and the target signal through pulse compression was obtained in [7]. Although this method had achieved the purpose of

anti-jamming, its realization in actual engineering remains to be studied. The idea of this type of method is to separate the target signal from mixed signal, and the difficulty lies in the separation algorithm.

The second type is to reconstruct the jamming signal, and then eliminate the jamming signal to obtain the target signal. For example, in [8], the parameters of the jamming signal were estimated according to the results of the short-time Fourier transform (STFT), and then the SMSP jamming from the mixed signal was reconstructed and suppressed. Although it can suppress jamming to a certain extent, it was only suitable for a certain jamming-to-noise ratio (JNR). An estimation method for different parameters of SMSP jamming was proposed and the SMSP jamming was reconstructed in [9], and then subtracted SMSP jamming from the mixed signal for suppression, but the jamming suppression effect was better only when the JNR was large. According to the idea of rapid segmented reconstruction of jamming signal and cancelled jamming by biorthogonal Fourier transform (BFT), the existing problem that the disturbed echo of the radar transmitting signal used as the processing object was solved in [10], but some coherent processing gain was lost. In [11], the SMSP jamming based on the estimated parameters was reconstructed by using the singular value (SVR) and SMSP jamming was subtracted from the received signal to suppress the jamming. This method had good robustness for sampling frequency. However, the better anti-jamming effect can be obtained only when the jamming-signal ratio (JSR) was large. This kind of idea is used to reconstruct the jamming signal, and then subtract the reconstructed jamming signal from the received echo to achieve the purpose of jamming suppression. The difficulty lies in the reconstruction method of the jamming signal. Although the proposed method can achieve jamming suppression, there are certain requirements for JSR.

The third type is multi-domain signal processing, which suppresses jamming based on the prominent difference between the target signal and jamming signal in a certain domain. For example, the time-frequency characteristic of the jamming signal and the target signal was used in [12]. The maximum entropy method (MEM) and genetic algorithm (GA) were used to obtain the segmentation threshold of the time-frequency filter, and the constructed time-frequency filter was used to suppress SMSP jamming. However, the segmentation threshold can only satisfy the constraints of the optimal solution, and it may not necessarily find the optimal solution. In [13], the SMSP jamming in the time-frequency domain was discarded, the time-frequency points belonging to the target signal were retained, a solution model for the CS minimum problem was established, and the orthogonal matching pursuit method was used to reconstruct the target signal to achieve the purpose of SMSP jamming suppression in [13]. Although this method can completely suppress SMSP jamming and effectively retain the target signal information, it cannot be implemented in real time. This kind of idea is to transform the received signal into other domains that can highlight the difference between the target signal and jamming signal, so as to achieve the purpose of jamming suppression. The difficulty lies in suppressing the jamming and recovering the target signal, because it will cause the loss of the target signal. Although the existing references can completely suppress the jamming, it is not necessarily optimal. There is a certain target signal loss, and the actual project cannot be realized quickly in real time.

Based on the background of self-defense jamming, LFM pulse compression radar suppresses SMSP jamming. We summarize the information from the abovementioned references and classify them in Table 1. Compared with the first method, the third method has the advantage of engineering realization, and it is not limited by the JSR, compared with the second method. Therefore, for the third method, this paper takes advantage of the difference between SMSP jamming and target signal in the time-frequency domain. In order to overcome the shortcomings that cannot be implemented quickly in actual engineering, a cutting compensation method for SMSP jamming suppression is proposed in the time-frequency domain. Because some time-frequency regions of the target signal overlap with SMSP jamming, which will cause the loss of the target signal, the target signal is compensated after the jamming suppressed. According to the inverse FRFT (IFRFT), the original coordinate system is restored to obtain the cancelled signal. The matched filter

can be used to check whether the jamming is effectively suppressed or not. At the same time, the changes of phase compensation and the changes of output signal to jamming plus noise ratio (SJNR) with the number of sub-pulses are analyzed in the presence of Doppler frequency. The time-frequency characteristic of SMSP jamming is analyzed, and the principles and specific implementation methods of jamming suppression are explained. Compared with the existing methods, it has the advantages of fast, real-time, and practical. Simulation results verify the feasibility of the algorithm.

**Table 1.** summarize and classify.

| Number | Method Ideas | Transform Domain or Signal Processing Tools |
|--------|--------------|---------------------------------------------|
| 1. | Separate the target signal and jamming | Polarization information [5] CS [6] BSS [5,7] |
| 2. | Reconstruct the jamming | STFT [8,9] BFT [10] SVR [11] |
| 3. | Multi-domain signal processing | MEM [12] GA [12] CS [13] |

## 2. SMSP Jamming Model

Without considering the echo delay factor and under the condition of self-defense jamming, assume that the radar transmitting signal is an LFM signal, which can be expressed as [6]

$$s(t) = A \cdot rect(t/T_p) \cdot e^{j(2\pi f_c t + \pi \mu t^2)}, \tag{1}$$

where $A$ is the complex envelope of the signal, $T_P$ is the pulse width, $\mu = B/T_P$ denotes modulation slope, $B$ is the bandwidth, $f_c$ is the initial frequency of the signal and $rect(t/T_p) = \begin{cases} 1, -T_p/2 \leq t \leq T_p/2 \\ 0, others \end{cases}$.

SMSP jamming is formed by the jammer compressing the LFM signal transmitted by the radar in the time domain, and then repeating it many times to keep the signal time width unchanged. The modulation slope of each sub-pulse is several times the target echo, so the signal sub-pulse of the SMSP jamming model generated by the jammer can be expressed as

$$j_c(t) = A_c \cdot rect\left(\frac{t}{T_p/k}\right) \cdot e^{j(2\pi f_c t + \pi k \mu t^2)}, \tag{2}$$

where $A_c$ is the complex envelope of the SMSP jamming, $k\mu$ is the frequency modulation slope of the SMSP jamming, and $k$ represents the number of sub-pulses. Because the sub-pulse repeats $k$ times, and the single SMSP jamming model can be expressed as

$$\begin{aligned} j(t) &= j_c(t) \otimes \sum_{i=1}^{k} \delta(t - iT_p/k) \\ &= \sum_{i=1}^{k} A_c \cdot rect\left(\frac{t - iT_p/k}{T_p/k}\right) \cdot e^{j\left(2\pi f_c(t - iT_p/k) + \pi k \mu(t - iT_p/k)^2\right)}, \end{aligned} \tag{3}$$

where $i = 1, 2 \cdots k$, $\otimes$ denotes the kronecker product, and $\sum$ denotes the sum symbol. According to (3), the phase of the SMSP jamming can be obtained as

$$\phi(t) = \sum_{i=1}^{k} rect\left(\frac{t - iT_p/k}{T_p/k}\right) \left[2\pi f_c(t - \frac{iT_p}{k}) + \pi k \mu(t - \frac{iT_p}{k})^2\right], \tag{4}$$

and the instantaneous frequency of the SMSP jamming can be written as [14]

$$
\begin{aligned}
F(t) &= \sum_{i=1}^{k} rect\left(\frac{t - iT_p/k}{T_p/k}\right) \cdot \left[f_c + k\mu(t - iT_p/k)\right] \\
&= \sum_{i=1}^{k} rect\left(\frac{t - iT_p/k}{T_p/k}\right) \cdot (k\mu t - iB + f_c).
\end{aligned}
\tag{5}
$$

It can be seen from (5) that the instantaneous frequency of the SMSP jamming is composed of $k$ lines with the same slope, and the slope of each line is $k\mu$, the intercept distance is $-iB + f_c$, and the sub-pulse width is $T_p/k$.

According to (3), the spectrum of the single SMSP jamming can be expressed as [15]

$$
\begin{aligned}
J(f) &= J_c(f) \cdot \sum_{i=1}^{k} e^{-j2\pi f \frac{i}{k}T_p} \\
&= A_c \sqrt{\frac{k}{\mu}} rect\left(\frac{f - f_c}{B}\right) e^{-j\pi \frac{(f-f_c)^2}{\mu}} e^{j\frac{\pi}{4}} \cdot e^{-j\pi f(1-\frac{1}{k})T_p} \frac{\sin c(fT_p)}{\sin c(fT_p/k)},
\end{aligned}
\tag{6}
$$

where $J_c(f)$ denotes the spectrum of the sub-pulse of the SMSP jamming.

According to the above analysis, it can be seen that the SMSP jamming and target signal completely overlap in the time domain and frequency domain, and the amplitude of the SMSP jamming is superimposed on the target signal, so we cannot suppress SMSP jamming in the time or frequency domain alone.

### 3. Suppressing SMSP Jamming in the Time-Frequency Domain
*3.1. FRFT*

FRFT is a time-frequency analysis tool, which has good energy accumulation characteristic for LFM signal.

The FRFT for LFM signal can be expressed as [16,17]

$$
\begin{aligned}
S(u) &= \int_{-\infty}^{\infty} K_p(t, u) s(t) dt \\
&= \sqrt{\frac{1 - j\cot\alpha}{2\pi}} \int_{-\infty}^{\infty} e^{j(\frac{u^2+t^2}{2}\cdot\cot\alpha - ut\csc\alpha)} s(t) dt,
\end{aligned}
\tag{7}
$$

where $u$ is the parameter of FRFT, which corresponds to $u$-domain. $K_p(t, u) = \sqrt{\frac{1-j\cot\alpha}{2\pi}} e^{j(\frac{u^2+t^2}{2}\cdot\cot\alpha - ut\csc\alpha)}$ is the kernel function of FRFT. $\alpha = p\frac{\pi}{2}$ is the rotation angle of FRFT ($\alpha \neq n\pi$, $n$ is an integer), $p$ is the order of FRFT, and $0 < |p| < 2$, so $0 < |\alpha| < \pi$. Owing to the frequency modulation slope of the transmitting signal is a priori knowledge as far as the radar receiver is concerned, we can compute the $p_\mu$th order FRFT $S_{p_\mu}(u)$ of $s(t)$ with $\alpha_\mu = arc\cot(-\mu)$ and $p_\mu = \frac{2}{\pi}\alpha_\mu$, where $arc$ denotes the inverse of the trigonometric function [18]. The $p_\mu$th order FRFT of the LFM signal can be expressed as follows. The derivation processes of (8) and (9) are shown in Appendix A.

$$
S_{p_\mu}(u) = AT_p\sqrt{\frac{1 + j\mu}{2\pi}} \cdot \sin c\left[\pi(-f_c + u\cdot\csc\alpha_\mu)T_p\right] \cdot e^{-j\pi\mu\cdot u^2}.
\tag{8}
$$

The $p_{k\mu}$th order FRFT of the SMSP jamming can be expressed as

$$
\begin{aligned}
J_{p_{k\mu}}(u) = A_c\frac{T_p}{k}\sqrt{\frac{1 + jk\mu}{2\pi}} \\
\cdot \left\{\sum_{i=1}^{k} e^{-j\pi\frac{i^2}{k}BT_p - j2\pi B\csc\alpha_{k\mu}\frac{i}{k}} \sin c\left[\pi(-f_c + iB + u\csc\alpha_{k\mu})\frac{T_p}{k}\right]\right\} e^{-j\pi k\mu\cdot u^2}.
\end{aligned}
\tag{9}
$$

According to (8) and (9), the FRFTs of the LFM signal emitted by radar and SMSP jamming are sinc functions, which have the effect of energy aggregation in the $u$-domain. However, the aggregation effect of the SMSP jamming is not as good as that of LFM signal. As shown in Figure 1, for the FRFT of the SMSP jamming, when the number of sub-pulses is larger, the sinc functions of sub-pulses will overlap with each other in the $u$-domain.

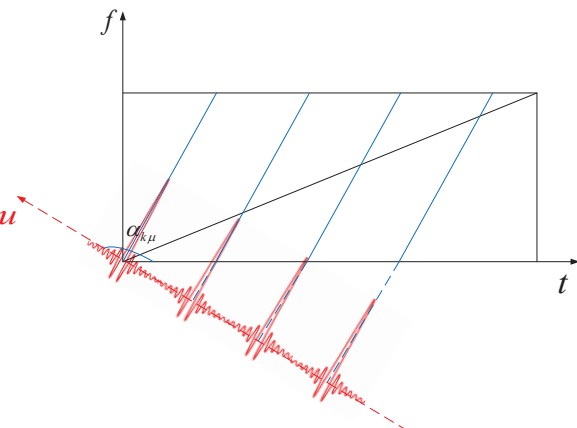

**Figure 1.** Schematic diagram of FRFT of SMSP jamming.

### 3.2. Cutting Compensation Method Based on Time-Frequency Domain

According to the characteristic of the FRFTs of the SMSP jamming and LFM signal transmitted by radar, the following method is proposed to suppress SMSP jamming in the time-frequency domain.

The pulse Doppler radar transmits LFM signal, and the mixed signal received by radar can be expressed as

$$y(t) = s(t - \tau_0) + j(t - \tau_j) + n(t), \tag{10}$$

where $\tau_0$ denotes the time delay of the target signal, $\tau_j$ denotes the time delay of the SMSP jamming, $n(t)$ represents Gaussian white noise. The FRFT of (10) is

$$\begin{aligned}
Y(u) &= \int_{-\infty}^{\infty} K_p(t, u) y(t) dt \\
&= \sqrt{\frac{1 - \mathrm{j}\cot\alpha}{2\pi}} \int_{-\infty}^{\infty} e^{\mathrm{j}(\frac{u^2 + t^2}{2} \cdot \cot\alpha - ut\csc\alpha)} \\
&\quad \cdot [s(t - \tau_0) + \mathrm{j}(t - \tau_j) + n(t)] dt.
\end{aligned} \tag{11}$$

Point 1: The amplitude of the jamming position in the $u$-domain is set to zero.

Two-dimensional peak search is carried out by using the $u$-domain and rotation angle $\alpha$ as unknown variables. According to (11), the maximum value of the function and the corresponding optimal rotation angle can be obtained. Compared with the power of the target signal and noise, the power of the SMSP jamming is larger, and the corresponding rotation angle is mainly the energy aggregation of the SMSP jamming. Assume that the optimal rotation angle is $\alpha_{k\mu}$, the corresponding amplitude is $Y_{p_{k\mu}}(u)$, and the intercepted amplitude is $Y_1(u)$. If $Y_{p_{k\mu}}(u) \geq Y_1(u)$, the $u$-domain units corresponding to energy accumulation of the SMSP jamming can be obtained, set as $u_{k\mu}$. We set

$$Y_{p_{k\mu}}\left(u_{k\mu}\right) = 0, \tag{12}$$

and the expression of $Y_{p_{k\mu}}(u)$ can change into $Y_r(u)$. Let $\alpha = -\alpha_{k\mu}$, and the IFRFT is performed on $Y_r(u)$ to obtain the representation of residual signal in the time domain, which can be denoted as $y_r(t)$. Equation (12) denotes that the jamming signal is set to zero

in the $u$-domain. This method can suppress the SMSP jamming completely, but a part of target signal is also suppressed, so the target signal needs to be compensated.

Point 2: The phase compensation of the target signal is considered.

The transmitting signal is known as far as the radar receiver is concerned, and the transmitting signal is transformed by the $p_{k\mu}$th order FRFT,

$$S_{p_{k\mu}}(u) = \sqrt{\frac{1 - j\alpha_{k\mu}}{2\pi}} \int_{-\infty}^{\infty} e^{j2\pi(\frac{u^2+t^2}{2} \cdot \cot \alpha_{k\mu} - ut \csc \alpha_{k\mu})} s(t) dt. \tag{13}$$

Equation (13) is equivalent to the rotation of the transmitting signal to the rotation angle $\alpha_{k\mu}$, containing phase information, which can be used to satisfy the phase compensation in the $u$-domain.

Point 3: The amplitude compensation of the target signal is considered.

According to (9), it can be known that the $u$-domain units corresponding to the energy accumulation of jamming are discontinuous distribution of several units. Because there are $k$ small pulses in the SMSP jamming, there will be $k$ peaks in the $u$-domain after FRFT, the $u$-domain unit $u_{k\mu}$ corresponding to the zero contains $k$ segments.

It is assumed that the segment $i$ in the $u$-domain is $u_{k\mu_i}$, and the units $u_{k\mu_i-}$ and $u_{k\mu_i+}$ are on the left and right sides of $u_{k\mu_i}$, the amplitude values on the left and right sides $Y_r(u_{k\mu_i-}) \neq 0$ and $Y_r(u_{k\mu_i+}) \neq 0$, the amplitude of compensation can be expressed as

$$|Y_r(u_{k\mu_i})| = (E[Y_r(u_{k\mu_i-})] + E[Y_r(u_{k\mu_i+})])/2, \tag{14}$$

where $E[\cdot]$ represents the mean value and $|\cdot|$ represents the modulus.

Point 4: The signal form after target signal compensation.

The compensation position of the target signal in segment $i$ can be expressed as

$$B(u) = |Y_r(u_{k\mu_i})| \frac{S_{p_{k\mu}}(u)}{|S_{p_{k\mu}}(u)|}. \tag{15}$$

After the target signal compensated, $Y_r(u)$ becomes $Y_n(u)$, and $Y_n(u)$ can be IFRFT to time domain. The obtained signal not only contains no SMSP jamming, but also has the same time-frequency characteristic as the transmitting signal.

*3.3. Processing of Cutting Compensation For SMSP Jamming Suppression*

In order to suppress SMSP jamming, the position of SMSP jamming in the $u$-domain should be estimated first. The transmitting signal is a priori knowledge as far as the radar receiver is concerned, and the rotation angle $\alpha_\mu$ can be obtained by FRFT to realize the $p_\mu$th order FRFT of the LFM signal. The rotation angle $\alpha_{k\mu}$ obtained by FRFT of the mixed signal can be used to obtain the jamming position because the power of SMSP jamming is much larger than that of the target signal and noise. The flowchart of the SMSP jamming suppression algorithm is shown in Figure 2.

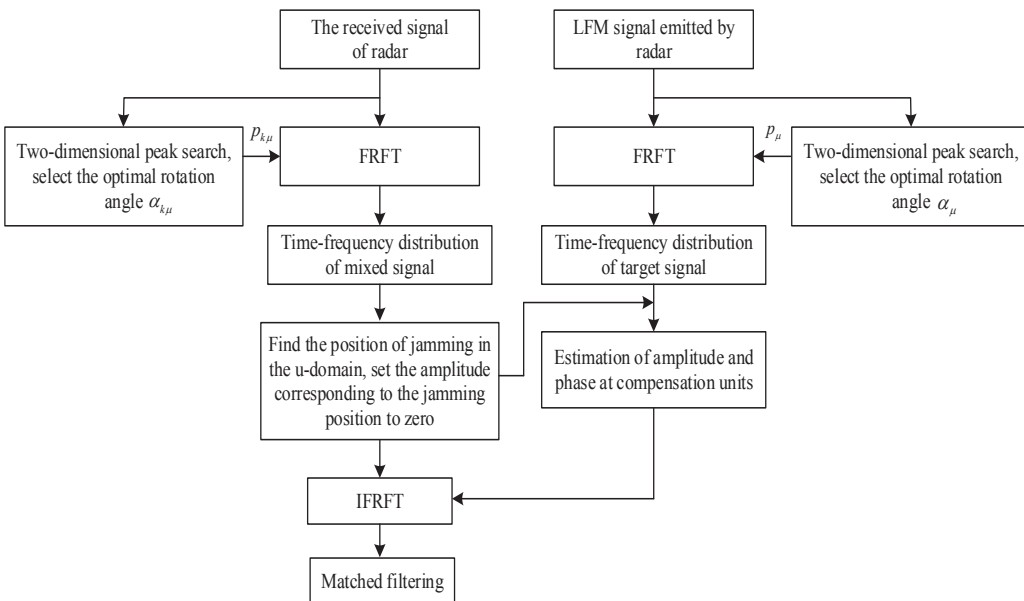

**Figure 2.** Flow diagram of the algorithm for SMSP jamming suppression.

Step 1: Through the FRFT of the ergodic rotation angle $\alpha$ of the receiving mixed signal and the transmitting LFM signal of the radar, the optimal rotation angles $\alpha_{k\mu}$ and $\alpha_{\mu}$ corresponding to the peak can be found by the two-dimensional peak search.

Step 2: The FRFT of the mixed signal and LFM signal at the optimal rotation angle $\alpha_{k\mu}$ and $\alpha_{\mu}$ are performed to obtain the characteristic of the mixed signal and LFM signal in the $u$-domain, which are corresponding to (11) and (13).

Step 3: In the $u$-domain, the intercept amplitude is set as $Y_1(u) = E[Y(u)] \times a$, where $a$ is the normalized threshold factor of the adaptive intercept, and it can be adaptively chosen by constant false alarm rate (CFAR). Find the position of the $u$-domain units $u_{k\mu} = find(u(Y_{k\mu}(u) > Y_1(u)))$ corresponding to the SMSP jamming, and then set the amplitude of the $u_{k\mu}$ units to zero, which is corresponding to (12).

Step 4: According to the time-frequency characteristic of the residual signal, the target signal is compensated. In the $u$-domain, the compensation position of the target signal are represented by $B(u)$, which is corresponding to (15).

Step 5: The time domain signal is obtained by the $p_{k\mu}$th order IFRFT of the compensated signal, and then match filtering for time domain signal.

## 4. Algorithm Analysis

### 4.1. Estimation of Doppler Frequency $f_d$

When there is a Doppler frequency, the received target signal is

$$s_r(t) = A \cdot rect\left(\frac{t - \tau_0}{T_p}\right) e^{j\left(2\pi(f_c + f_d)(t - \tau_0) + \pi\mu(t - \tau_0)^2\right)}. \tag{16}$$

The SMSP jamming signal can be expressed as

$$j_r(t) = \sum_{i=1}^{k} A_c \cdot rect\left(\frac{t - \tau_j - iT_p/k}{T_p/k}\right) \cdot e^{j\Phi}, \tag{17}$$

where $\Phi = 2\pi(f_c + f_d)(t - \tau_j - iT_p/k) + \pi k\mu(t - \tau_j - iT_p/k)^2$.

Assume that the starting points of the FRFT are the same, which is aligned on the range gate, the Doppler frequency can be estimated and the phase of the received signal

can be compensated. In the presence of Doppler frequency, the $p_\mu$th order FRFT of the residual signal $y_r(t)$ is transformed into

$$Y_{p_\mu}(u) = \sqrt{\frac{1-\mathrm{j}\cot\alpha_\mu}{2\pi}} \int_{-\infty}^{\infty} e^{\mathrm{j}2\pi(\frac{u^2+t^2}{2}\cdot\cot\alpha_\mu - ut\csc\alpha_\mu)}y_r(t)dt. \tag{18}$$

Although $y_r(t)$ does not contain the SMSP jamming, a part of the target signal is lost. The maximum amplitude after the $p_\mu$th order FRFT is larger than the $p_\mu$th order FRFT when there is only the target signal. That is, compared with (8), the maximum amplitude of (18) is reduced. Considering the Doppler frequency, the $u$-domain units of the target frequency will shift, as shown in Figure 3.

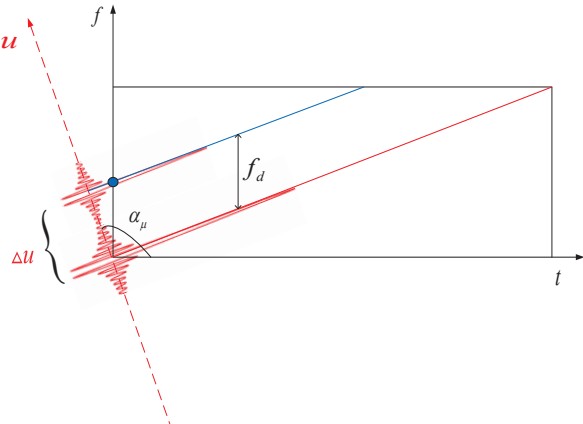

**Figure 3.** Rotation angle relation.

According to Figure 3, the relationship among rotation angle $\alpha_\mu$, the Doppler frequency $f_d$, and offset $\Delta u$ of the $u$-domain units corresponding to the target signal caused by the Doppler frequency can be analyzed:

$$f_d = \Delta u / \sin(\pi - \alpha_\mu) = \Delta u / \sin(\pi - p_\mu \pi/2). \tag{19}$$

Thus, when there is a Doppler frequency, (15) can be changed into

$$B(u) = |Y_r(u_{k\mu_i})|\frac{S_{p_{k\mu}}(u)}{|S_{p_{k\mu}}(u)|}e^{-\mathrm{j}2\pi f_d t}. \tag{20}$$

However, the error problems such as range gate misalignment, spectral width, and pulse leading edge will be considered in practice, the frequency difference $\Delta f_d$ is uniformly expressed here, and the phase also needs to be compensated. Therefore, replace $e^{-\mathrm{j}2\pi(f_d+\Delta f_d)t}$ with $e^{-\mathrm{j}2\pi f_d t}$ in (20).

### 4.2. Adaptive Selection of Cutting Threshold

In actual radar signal processing, the statistical characteristic of noise and jamming is rarely known in advance, and usually variable, so it is difficult to calculate a fixed threshold in advance. In fact, the statistical characteristic of jamming can be estimated from the received data, and then the threshold is calculated by the statistical characteristic of SMSP jamming. Therefore, CFAR is used for the selection of $a$ in the Section 3.3 [19,20], as shown in Figure 4.

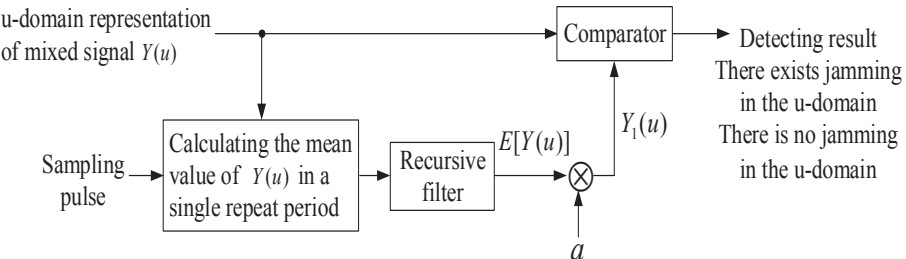

**Figure 4.** Adaptive threshold CFAR processing in the time-frequency domain.

The recursive filter outputs are the mean estimation of the input signal in the $u$-domain, multiplied by the normalized threshold factor $a$ to obtain the adaptive threshold $Y_1(u) = E[Y(u)] \times a$. $a$ should be adjusted according to the jamming intensity and the number of distance units involved in the average calculation.

*4.3. Output SJNR*

In order to conveniently represent the power comparison among the target signal, SMSP jamming and noise, the SJR of the received signal can be expressed as

$$SJR = \frac{A^2 E[|s_r(t)|^2]}{A_c^2 E[|j_r(t)|^2]}, \tag{21}$$

where the SMSP jamming can be seen as $k$ sub-pulses, the modulation slope is $k$ times the target signal, $j_r(t) = kj_{r_i}(t)$. At the time of suppressing the SMSP jamming in time-frequency domain, a part of target signal is also suppressed. The time-frequency characteristic of the target signal is equivalent to be divided into $k$ segments in the time-frequency domain, and the intermediate discontinuous part is equivalent to the loss of the target, as shown in Figure 5. Assume that the target signal lost at each discontinuity is $z_i(t)$, $i = 1, 2, \cdots, k$, the amplitude is $A_i$ and the SMSP jamming is completely suppressed, so the output SJNR can be expressed as

$$SJNR = \frac{A^2 E[|s_r(t)|^2 - \sum\limits_{i=1}^{k} E[A_i^2|z_i(t)|^2]}{E[|n_1(t)|^2]}, \tag{22}$$

where $n_1(t)$ is the noise after the cutting operation. It can be seen from (22) that the larger $k$, the more loss of the target signal. That is, the bigger the modulation slope of SMSP jamming sub-pulse, the more loss of the target signal.

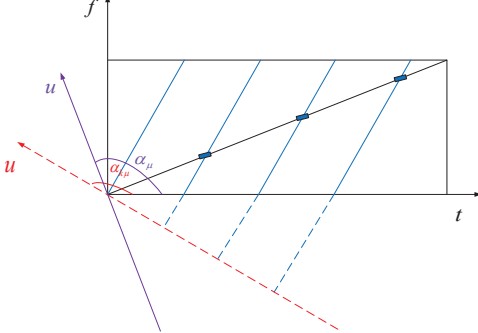

**Figure 5.** Principle diagram of SMSP jamming suppression in the time-frequency domain.

**5. Simulation and Analysis**

The proposed SMSP jamming suppression algorithm is simulated according to the simulation parameters in Table 2.

**Table 2.** Simulation parameters.

| | |
|---|---|
| Initial frequency (MHz) | 4.5 |
| Bandwidth (MHz) | 5 |
| pulse width (us) | 100 |
| Frequency modulation slope (GHz/s) | 50 |
| Sampling rate (MHz) | 10 |
| The sub-pulse width of SMSP jamming (us) | 10 |
| The number of sub-pulses | 10 |
| Frequency modulation slope of the sub-pulse (GHz/s) | 500 |
| SJR (dB) | −16.9 |
| The Doppler frequency (MHz) | 0.5 |

Figure 6 shows the time-frequency distribution characteristic of the LFM signal transmitted by the radar. It can be seen that the frequency of the LFM signal increases with time. Figure 7 shows the time-frequency distribution of the mixed signal received by the receiver. It can be found that the SMSP jamming is mixed with the LFM signal in the frequency domain, so the SMSP jamming cannot be distinguished and suppressed in the frequency domain. Figure 8 shows the matched filtering result of the echo signal received by the radar. Although the target signal can be well matched filtering in the time domain, the power of the jamming is too large, causing the target signal and SMSP jamming to be unable to be distinguished in the time domain.

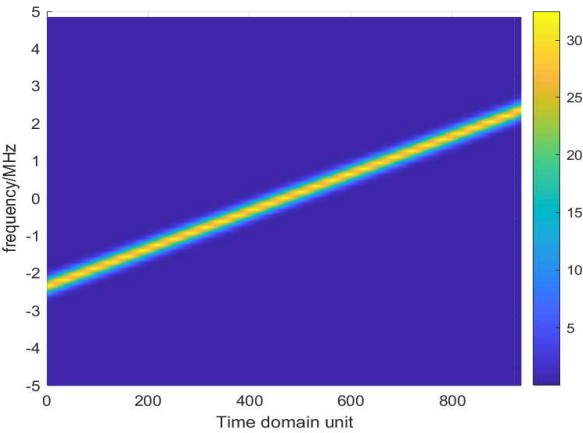

**Figure 6.** Time-frequency distribution of the transmitting LFM signal.

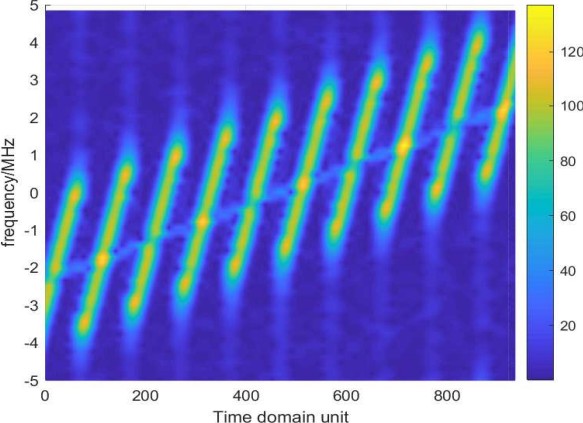

**Figure 7.** Time-frequency distribution of the mixed signal.

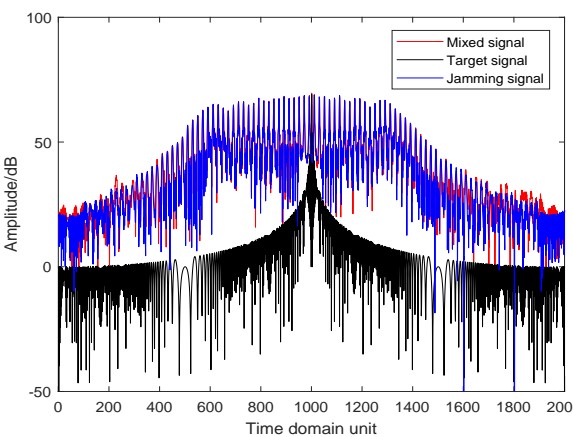

**Figure 8.** Pulse compression result of the output signal.

Figure 9 represents the three-dimensional diagram of the FRFT of the receiving mixed signal, and the two-dimensional peak search is carried out in the time-frequency domain. Due to the large power of the SMSP jamming, it can be concluded that the order corresponding to the maximum SMSP jamming is $p_{k\mu} = 1.88$th order, so that the mixed signal can be rotated to the $u$-domain with the rotation angle $\alpha_{k\mu} = 1.88 \times \frac{\pi}{2}$.

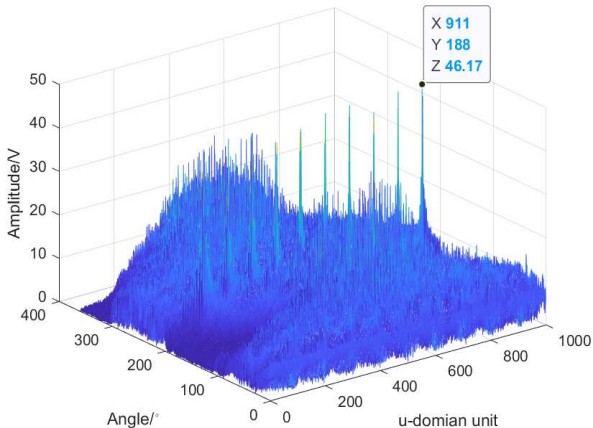

**Figure 9.** FRFT of the mixed signal.

Figure 10 shows the result of the mixed signal and the target signal after the $p_{k\mu} = 1.88$th order FRFT. Because the power of the SMSP jamming is much greater than the target signal, the peak position is the corresponding position of the SMSP jamming in the $u$-domain. For detection in the simulation, set $a = 0.9$, that is, set the amplitude of the mixed signal to 0.9 times the average amplitude of the mixed signal for interception, and obtain the residual signal in the $u$-domain, as shown in Figure 11.

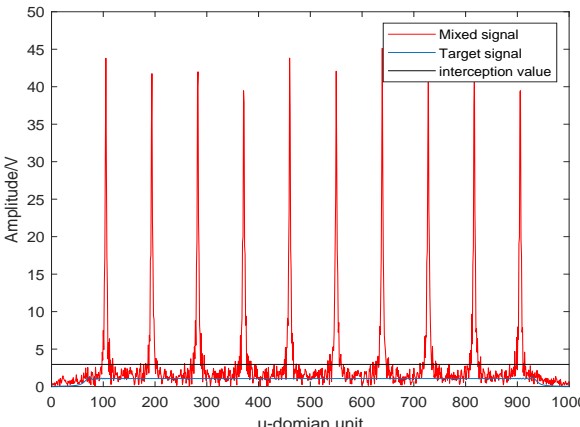

**Figure 10.** *u*-domain waveform of mixed signal.

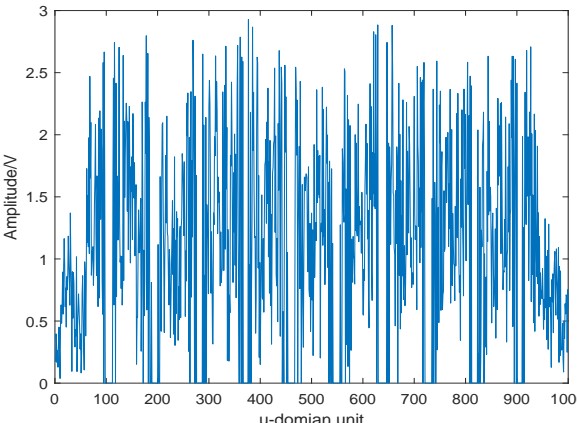

**Figure 11.** Cross-section diagram of *u*-domain waveform of mixed signal.

Figure 12 shows the time-frequency distribution characteristic of the residual signal after jamming was suppressed. Because a part of the target signal overlaps with the SMSP jamming in the time-frequency domain, the target signal is also lost while the jamming is completely suppressed, so the time-frequency distribution of the residual target signal is discontinuous. In the *u*-domain, the signal is compensated at the points where the SMSP jamming set to zero, and the compensated *u*-domain waveform is shown in Figure 13.

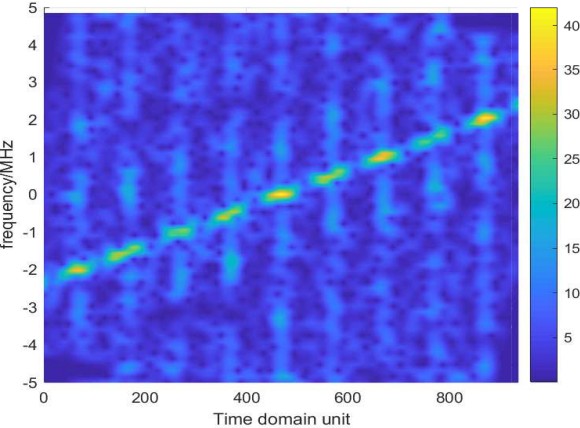

**Figure 12.** Time-frequency distribution of the residual signal after jamming suppressed.

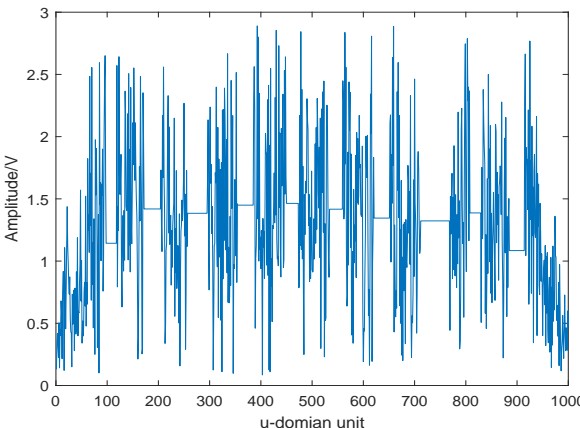

**Figure 13.** *u*-domain waveform of the compensated signal.

Figure 14 shows the time-frequency distribution of the target signal obtained by the $p_{k\mu}$th order IFRFT after the SMSP jamming suppressed in the *u*-domain. It can be seen that the time-frequency points of the target signal are still retained, which is consistent with the time-frequency distribution of the LFM signal transmitted by the radar in Figure 6. Figure 15 is obtained by matched filtering of the recovered time domain signal. It can be seen that the SMSP jamming is effectively suppressed, and the obtained peak of the target signal has no sidelobe. This shows that this method can effectively suppress SMSP jamming and extract the target signal.

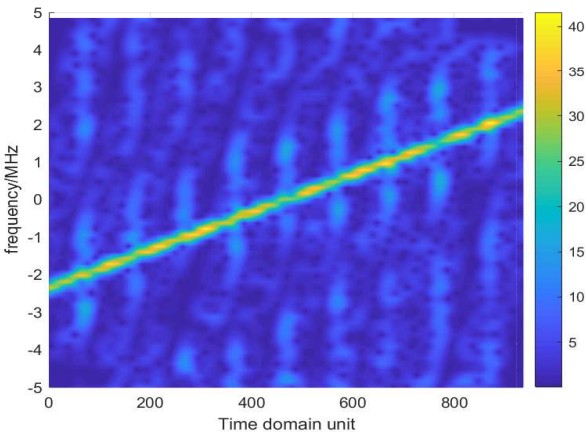

**Figure 14.** Time-frequency distribution of the compensated signal.

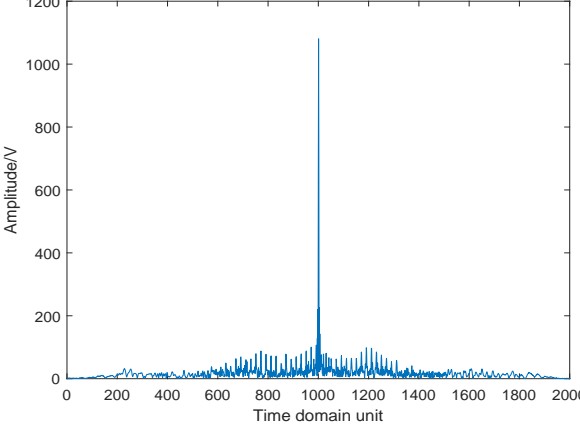

**Figure 15.** Matched filtering output of the compensated signal.

The following is the verification of the algorithm in the presence of Doppler frequency. Figure 16 shows the time-frequency distribution of the mixed signal with Doppler frequency. Compared with Figure 7 without Doppler frequency, the frequency of the mixed signal is shifted upward. Figure 17 shows the $u$-domain waveforms of the residual signal and the transmitting LFM signal after SMSP jamming was suppressed. Due to the presence of the Doppler frequency, the target signal has an offset in the $u$-domain. The Doppler frequency can be obtained by using the relationship among the offset, rotation angle and the Doppler frequency.

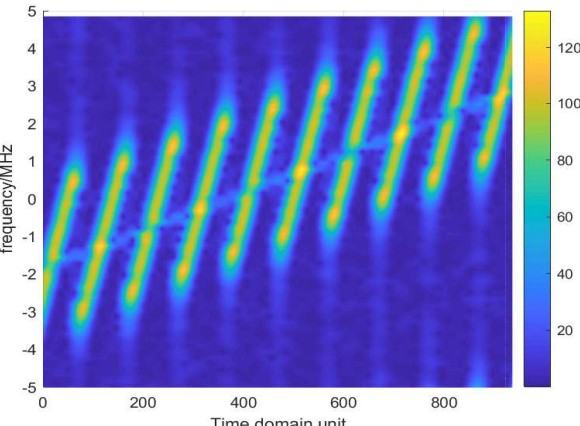

**Figure 16.** Time-frequency distribution of mixed signal when there is a Doppler frequency.

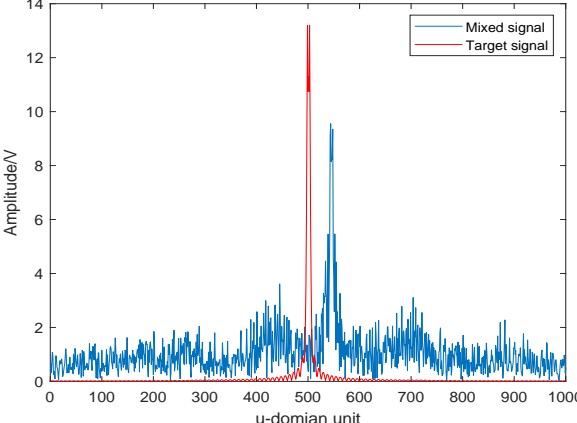

**Figure 17.** The $u$-domain waveform of the signal after jamming suppressed and the transmitting LFM signal at $\alpha_\mu$ when there is a Doppler frequency.

Figure 18 shows that the proposed method in this letter can be used to suppress SMSP jamming, and the time-frequency distribution of the signal after the Doppler frequency compensation is consistent with that of the LFM signal transmitted by the radar in Figure 6. Figure 19 shows the matched filtering of the time domain signal with Doppler frequency. It can be also proved that the proposed method can effectively suppress SMSP jamming.

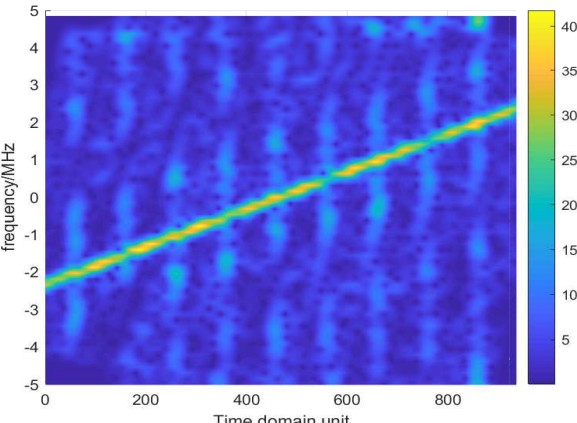

**Figure 18.** Time-frequency distribution of compensated signal when there is a Doppler frequency.

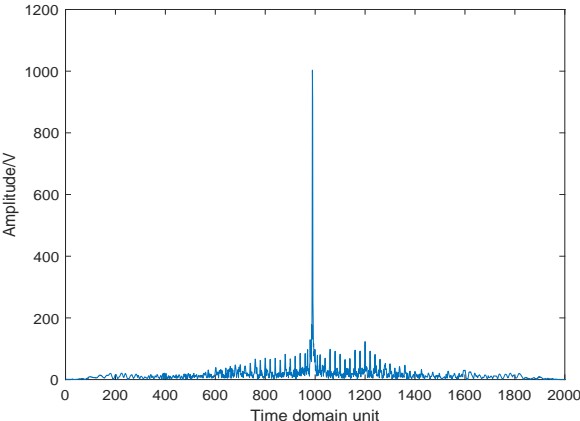

**Figure 19.** Time-frequency distribution of residual signal when there is a Doppler frequency.

Figure 20 shows the variation of the output SJNR with the number of SMSP jamming sub-pulses. It can be seen that the output SJNR decreases as the number of SMSP jamming sub-pulses increases after the signal compensated. The more the number of sub-pulses, the greater the modulation slope and the greater the loss of the target signal, corresponding to (22). At the same time, it can be seen that the output SJNR without compensation increases and then slowly decreases with the number of sub-pulses, which is mainly due to (22). When the numerator and denominator decrease at the same time and the number of sub-pulses is small, the denominator decreases more than the numerator. As the number of sub-pulses increases, the power loss of the residual target signal is greater, so the output SJNR increases and then decreases. At the same time, it also can be seen that the output SJNR decreases as the input SJR decreases, and when the input SJR is small enough, the output SJNR without compensation always decreases as the number of sub-pulses increases.

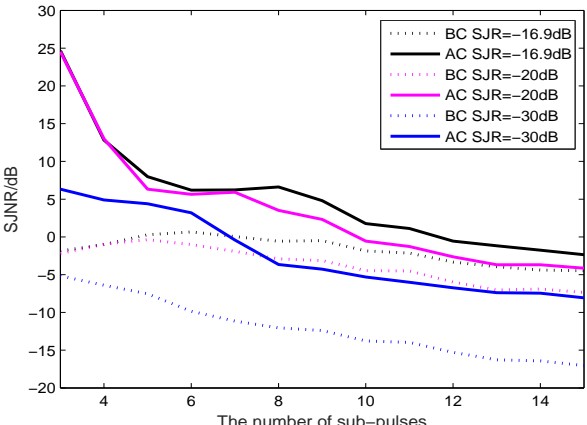

**Figure 20.** The curve of output SJNR with the number of sub-pulses. In the legend, "BC" denotes "before compensation", while "AC" denotes "after compensation".

## 6. Conclusions

By analyzing the distribution characteristic of the SMSP jamming and LFM signal in the time-frequency domain, a method of identification and suppression was proposed based on FRFT. This method completely suppresses SMSP jamming in the time-frequency domain, and uses the compensation method to restore the time-frequency distribution characteristic of the target signal. The simulation results show the correctness of the theoretical analysis and effective anti-jamming of the proposed method. It has the advantages of fast, real-time, practical, etc., and provides reference value for project implementation. In the future, jamming suppression algorithms and unity problems under different jamming background will be studied.

**Author Contributions:** Conceptualization, H.C. and W.L.; methodology, L.Z. and Z.Z.; validation, Z.Z. and H.C.; resources, W.L. and Y.W.; data curation, L.Z.; writing—original draft preparation, L.Z.; writing—review and editing, L.Z. and L.N.; supervision, Y.W. All authors have read and agreed to the published version of the manuscript.

**Funding:** This research was funded by the National Nature Science Foundation of China, grant number 62071482, 62101593 and 62001510.

**Conflicts of Interest:** The authors declare no conflict of interest.

## Appendix A

We can derive (8) from the following derivation

$$
\begin{aligned}
S_{p_\mu}(u) &= \sqrt{\frac{1 - \mathrm{j}\cot\alpha_\mu}{2\pi}} \int_{-\infty}^{\infty} e^{\mathrm{j}2\pi\left(\frac{u^2+t^2}{2}\cdot\cot\alpha_\mu - ut\csc\alpha_\mu\right)} s(t)dt \\
&= A\sqrt{\frac{1 - \mathrm{j}\cot\alpha_\mu}{2\pi}} e^{\mathrm{j}\pi u^2\cot\alpha_\mu} \int_{-\frac{T_p}{2}}^{\frac{T_p}{2}} e^{\mathrm{j}\pi(\cot\alpha_\mu + \mu)t^2} e^{\mathrm{j}2\pi(f_c - u\cdot\csc\alpha_\mu)t}dt \qquad \text{(A1)}\\
&= AT_p\sqrt{\frac{1 + \mathrm{j}\mu}{2\pi}} \cdot \sin c\left[\pi\left(-f_c + u\cdot\csc\alpha_\mu\right)T_p\right] \cdot e^{-\mathrm{j}\pi\mu\cdot u^2},
\end{aligned}
$$

and derive (9) from the following derivation

$$
\begin{aligned}
J_{p_{k\mu}}(u) &= \sqrt{\frac{1 - \mathrm{j}\cot\alpha_{k\mu}}{2\pi}} \cdot \int_{-\infty}^{\infty} e^{\mathrm{j}2\pi(\frac{u^2+t^2}{2}\cdot\cot\alpha_{k\mu} - ut\csc\alpha_{k\mu})} j(t)dt \\
&= A_c \sqrt{\frac{1 - \mathrm{j}\cot\alpha_{k\mu}}{2\pi}} \sum_{i=1}^{k} \int_{\frac{2i-1}{2k}T_p}^{\frac{2i+1}{2k}T_p} e^{\mathrm{j}(2\pi f_c(t - iT_p/k) + \pi k\mu(t - iT_p/k)^2)} e^{\mathrm{j}2\pi(\frac{u^2+t^2}{2}\cdot\cot\alpha_{k\mu} - ut\csc\alpha_{k\mu})} dt \\
&= A_c \sqrt{\frac{1 + \mathrm{j}k\mu}{2\pi}} e^{-\mathrm{j}\pi k\mu\cdot u^2} \cdot \sum_{i=1}^{k} e^{-\mathrm{j}2\pi f_c \frac{i}{k}T_p} \int_{\frac{2i-1}{2k}T_p}^{\frac{2i+1}{2k}T_p} e^{\mathrm{j}(2\pi f_c t + \pi k\mu(t - iT_p/k)^2)} e^{\mathrm{j}2\pi(-\frac{t^2}{2}\cdot k\mu - ut\csc\alpha_{k\mu})} dt \\
&= A_c \frac{T_p}{k} \sqrt{\frac{1 + \mathrm{j}k\mu}{2\pi}} \cdot \left\{ \sum_{i=1}^{k} e^{-\mathrm{j}\pi\frac{i^2}{k}BT_p - \mathrm{j}2\pi B\csc\alpha_{k\mu}\frac{i}{k}} \cdot \mathrm{sin}\,c\left[ \pi(-f_c + iB + u\csc\alpha_{k\mu})\frac{T_p}{k} \right] \right\} e^{-\mathrm{j}\pi k\mu\cdot u^2}.
\end{aligned}
\tag{A2}
$$

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
