# Peer review of "Cutting Compensation in the Time-Frequency Domain for Smeared Spectrum Jamming Suppression"

_electronics, doi:10.3390/electronics11131970_

Round 1

Reviewer 1 Report

This paper has demonstrated a fast Smeared Spectrum (SMSP) jamming suppression method based on the time-frequency domain analysis. The strengths:•       Problem is well defined and motivated.•       Discussion of related works is thorough.•       Figures are clear.•       References are up to date.

Including comments on the processing gain compared to other methods such as cancellation mentioned in the Introduction Section will improve the paper.

Author Response

Thank you for your careful reading, comments, and suggestions. Compared with the existing methods, our proposed method has the advantages of fast, real-time, and practical implement. The processing gain of the proposed method is related to the adaptive selection of the cutting threshold, and the selection of threshold is related to the surrounding environment of background condition. Hence, it may not be suitable to compare with other methods in processing gain.

Reviewer 2 Report

I read carefully this paper, and compared with the results presented in reference 8 (115 (2020). https://doi.org/10.1186/s13638-020-01728-y). However, I saw these results are very similar, and in the paper it is not clear where the novelty is and what is introduce new regarding reference 8. I suggest more clear explanation in the introduction part. Some additional issues should be considered by the authors:

  1. In page 4, Equation 1. Is this equation proposed by the authors or very well known?, in the second option a reference is needed.
  2. The same apply for equations 2,3,4
  3. Figures 1 and 3 are very similar. I suggest to unify as Fig a and b and if possible also to unify the description.
  4. There grammar mistakes. The manuscript should be revised carefully. For example in the caption of Figure 16 it is written Dopplor frequency, etc.
  5. In the conclusion part, the authors should mention the concrete application in which their results are advantageous and to mention future works and research to be done in that direction.

Author Response

General comment: I read carefully this paper, and compared with the results presented in reference 8 (115 (2020). https://doi.org/10.1186/s13638-020-01728-y). However, I saw these results are very similar, and in the paper it is not clear where the novelty is and what is introduce new regarding reference 8. I suggest more clear explanation in the introduction part. Some additional issues should be considered by the authors:

Response: Thank you for your comments. We have reshaped the paper to make the novelty more clearly. Compared with [8], we propose a method to eliminate the SMSP jamming in the time-frequency domain directly without reconstructing jamming. In contrast, the parameters of the jamming signal were estimated based on the STFT result in [8], and then the jamming signal was reconstructed and suppressed form the mixed signal. The proposed method in this paper has the advantages of fast, real-time, practical implement, etc. Moreover, we can obtain the Doppler frequency in the time-frequency domain, which can be applied in practical engineering applications.

Point 1: In page 4, Equation 1. Is this equation proposed by the authors or very well known? , in the second option a reference is needed.

Response: Many thanks for your careful reading. Equation 1 is well known, and a reference has been added, namely, reference [6]. The reference [6] has also been added in the second option.

Point 2: The same apply for equations 2, 3, 4

Response: Many thanks for your comments. Equations 2, 3, 4 are proposed by us, which can reflect the processing of generating SMSP jamming.

Point 3: Figures 1 and 3 are very similar. I suggest to unify as Fig a and b and if possible also to unify the description.

Response: Thank you for your suggestion. Figure 1 shows the characteristics of SMSP jamming in u-domain after FRFT without Doppler frequency, and the characteristics of the sinc function in the u-domain corresponding to equation (9). In contrast, Figure 3 shows Doppler frequency estimated according to the relationship between rotation angle and Doppler frequency when Doppler frequency. Hence, it may not be suitable to putting them together in one Figure.

Point 4: There grammar mistakes. The manuscript should be revised carefully. For example in the caption of Figure 16 it is written Dopplor frequency, etc.

Response: Many thanks for your careful reading. We have corrected Dopplor as Doppler and revised the manuscript carefully.

Point 5: In the conclusion part, the authors should mention the concrete application in which their results are advantageous and to mention future works and research to be done in that direction.

Response: Thank you for your valuable suggestion, which are very helpful for improving the quality of our manuscript. Aiming at the main lobe jamming, the proposed method uses the difference between the target signal and the SMSP jamming in the time-frequency domain to design the targeted jamming suppression method from the radar signal processing level, which provides technical reference for the anti-jamming design of the radar system. In the future, jamming suppression algorithms and unity problems under different jamming backgrounds will be studied. We have added the future research trick in the end of the revised manuscript.

In summary, we have modified the manuscript significantly, and we appreciate for reviewers’ warm comment earnestly.

Reviewer 3 Report

The paper is interesting and supported by good results even if the presentation can be improved.

The introduction is very detailed and well illustrate the objectives and the bases but  should be made more readable  (suggestion use less the word reference).

Perhaps a table to summarize the information from the papaers mentioned in the introduction and classify them more analytically would be positive.

Also mathematically, it would be useful to make the equation more readable.

In particular, for equations (8) and (9) it would be preferable to report only the final result and postpone the calculations to an annex

Paragraphs 3.2 and 3.3 are divided into steps (paragraph 3.2 are only numbered and the word step is lost) this somewhat damages the readability of the text.

A more fluent approach would be recommended.

Equation 21 reports JSR instead of SJR

In general, the paper needs to be made more readable

Author Response

General comment: The paper is interesting and supported by good results even if the presentation can be improved. The introduction is very detailed and well illustrate the objectives and the bases but should be made more readable (suggestion use less the word reference).

Response: Thank you for your encouraging and valuable comments. We have deleted the word reference in the revised manuscript.

Point 1: Perhaps a table to summarize the information from the papers mentioned in the introduction and classify them more analytically would be positive. Also mathematically, it would be useful to make the equation more readable. In particular, for equations (8) and (9) it would be preferable to report only the final result and postpone the calculations to an annex.

Response: Thank you for your valuable comments. We have made a table to summarize the information from the references mentioned in the introduction and classify them, i.e., Table 1. For equations (8) and (9), we have reported only the final results, and shifted the detailed derivations to the appendix.

Point 2: Paragraphs 3.2 and 3.3 are divided into steps (paragraph 3.2 are only numbered and the word step is lost) this somewhat damages the readability of the text. A more fluent approach would be recommended.

Response: Thank you for your suggestion. We have modified accordingly. Precisely, we have added some statements in Paragraphs 3.2 in the revised manuscript.

Point 3: Equation 21 reports JSR instead of SJR. In general, the paper needs to be made more readable.

Response: Many thanks for your careful reading. We have corrected this mistake.

In summary, we have modified the manuscript significantly, and we appreciate for reviewers’ warm comment earnestly.
